# DSPFusion: Degradation and Semantic Prior Dual-guided Framework for Image Fusion

## Abstract

Existing fusion methods are tailored for high-quality images but struggle with degraded images captured under harsh circumstances, thus limiting the practical potential of image fusion. In this work, we present a **D**egradation and **S**emantic **P**rior dual-guided framework for degraded image **Fusion** (**DSPFusion**), utilizing degradation priors and high-quality scene semantic priors restored via diffusion models to guide both information recovery and fusion in a unified model. In specific, it first individually extracts modality-specific degradation priors and jointly captures comprehensive low-quality semantic priors from cascaded source images. Subsequently, a diffusion model is developed to iteratively restore high-quality semantic priors in a compact latent space, enabling our method to be over $200\times$ faster than mainstream diffusion model-based image fusion schemes. Finally, the degradation priors and high-quality semantic priors are employed to guide information enhancement and aggregation via the dual-prior guidance and prior-guided fusion modules. Extensive experiments demonstrate that DSPFusion mitigates most typical degradations while integrating complementary context with minimal computational cost, greatly broadening the application scope of image fusion.

## 1 Introduction

Image fusion is a fundamental enhancement technique designed to combine complementary context from multiple images, thereby overcoming the limitations of single-modality or single-type sensors (Zhang et al., 2021). Infrared-visible image fusion (IVIF) is a key research area in image fusion, integrating essential thermal information from infrared (IR) images with the rich textures of visible (VI) images for comprehensive scene characterization (Zhang & Demiris, 2023). The complete information integration and visually pleasing results make IVIF widely applied in military detection (Muller & Narayanan, 2009), security surveillance (Zhang et al., 2018), assisted driving (Bao et al., 2023), object detection (Jain et al., 2023), semantic segmentation (Zhang et al., 2023), *etc.*

Recently, IVIF has garnered significant attention, leading to rapid advancements in relevant algorithms. These algorithms can be classified based on network architecture into convolutional neural network-based (Ma et al., 2021; Zhao et al., 2023a), autoencoder-based (Li & Wu, 2019; Li et al., 2023a), generative adversarial network-based (Ma et al., 2019; Liu et al., 2022), Transformer-based (Ma et al., 2022; Zhang et al., 2022), and diffusion model (DM)-based (Zhao et al., 2023b; Yi et al., 2024a) methods. Alternatively, from a functional perspective, these algorithms can be categorized into visual-oriented (Ma et al., 2019; Tang et al., 2022c), degradation-aware (Tang et al., 2023; Yi et al., 2024b; Zhang et al., 2024), semantic-driven (Tang et al., 2022b; Liu et al., 2023a), and joint registration-fusion (Tang et al., 2022a; Xu et al., 2023) approaches. Despite the satisfactory fusion performance achieved by these methods, several challenges still remain. On the one hand, while diffusion models with powerful generative abilities could bring gains, DM-based fusion methods (Zhao et al., 2023b; Yue et al., 2023) are often computationally intensive and time-consuming, making it unapplicable in real-time tasks such as assisted driving and security surveillance. On the other hand, although some degradation-aware methods have been proposed to address imaging interferences, they still struggle with complex fusion scenarios. For example, DIVFusion (Tang et al., 2023) and PAIF (Liu et al., 2023b) are tailored for specific degradations (*e.g.,* low-light or noise) but fail to generalize to others. Additionally, there are some general degradation-aware methods that handle multiple degradations within a single framework assisted by additional semantic context, such as text prompts (Yi et al., 2024b). However, they are sensitive to text prompts and struggle to han-

dle cases where degradations occur simultaneously in both infrared and visible images. Moreover, tailoring text descriptions for each fusion scenario is challenging.

To overcome the above challenges, we propose a degradation and semantic prior dual-guided image fusion framework, abbreviated as DSPFusion, which incorporates degradation suppression and information aggregation into a unified model without additional assistance. The proposed method involves two training phases. In Stage I, the semantic prior embedding network (SPEN) captures the semantic prior from cascaded high-quality sources, while the degradation prior embedding network (DPEN) extracts distinct degradation priors from two degraded images separately. A Transformer-based restoration and fusion network, guided by semantic and degradation priors via the dual prior guidance module, synthesizes high-quality fusion results. Note that the scene semantic prior is jointly derived from both modalities, enabling our model to enhance one using high-quality context from the complementary modality. A contrastive mechanism is employed to constrain the training of DPEN, thus ensuring the degradation priors effectively characterize various degradation types. Since high-quality source images are unavailable in practical situations, we deploy a diffusion model to restore high-quality semantic priors from low-quality ones in Stage II. The diffusion process is performed in a compact latent space, making our model computationally efficient and lightweight. Ultimately, DPEN adaptively identifies different degradation types and a diffusion model refines semantic priors, assisting the restoration and fusion model in synthesizing high-quality fusion results. In summary, our main contributions are as follows:

- We propose a novel restoration and fusion framework with dual guidance of degradation and semantic priors, effectively handling most typical degradations (*e.g.*, low-light, over-exposure, noise, blur, and low-contrast) while aggregating complementary information from multiple source images in one unified model. To our knowledge, it is the first model that comprehensively addressing various degradations in image fusion.

- A diffusion model is devised to restore high-quality semantic priors in a compact latent space, providing coarse-grained semantic guidance with low computational complexity.

- A contrastive mechanism is employed to constrain DPEN to adaptively perceive degradation types from source images, thereby guiding the restoration and fusion network as well as the semantic prior diffusion model to purposefully handle degradations without requiring additional auxiliary information.

- Extensive experiments on normal and degraded scenarios demonstrate the superiority of our method in degradation suppression and complementary context aggregation. Remarkably, it is two orders of magnitude more efficient than mainstream DM-based fusion algorithms.

## 2 RELATED WORK

**Image Fusion.** Earlier visual-oriented fusion methods focus on integrating cross-modal complementary context and enhancing visual quality, which rely on elaborate network architectures and loss functions to preserve complementary information that remains faithful to source images. Initially, mainstream network architectures primarily include CNN (Liang et al., 2022; Zhao et al., 2023a), AE (Li & Wu, 2019; Li et al., 2023a), and GAN (Ma et al., 2019; Liu et al., 2022). With the rise of Transformers (Vaswani et al., 2017) and diffusion models (Ho et al., 2020), these architectures gradually dominate fusion model design (Ma et al., 2022; Yue et al., 2023). However, as a compute-intensive process, the time cost of diffusion models remains a contentious issue.

Furthermore, several schemes, including joint registration and fusion (Xu et al., 2022b; Wang et al., 2022; Xu et al., 2023), semantic-driven (Tang et al., 2022b; Liu et al., 2022; Sun et al., 2022), and degradation-aware (Tang et al., 2023; Liu et al., 2023b) methods are proposed to broaden the practical applications of image fusion. Particularly, under some extreme conditions, environmental factors like low light and noise inevitably affect imaging. Thus, Tang et al. (2023) proposed an illumination-robust fusion method, achieving low-light enhancement and complementary context aggregation simultaneously. Liu et al. (2023b) developed a perception-aware method by leveraging adversarial attack and architecture search to boost the robustness of the fusion network and downstream tasks against noise. However, these methods are tailored to specific degradations and struggle with complex and diverse interferences. To this end, Yi et al. (2024b) leveraged CLIP to extract semantic embedding from text descriptions to assist the fusion network in addressing multiple degradations.

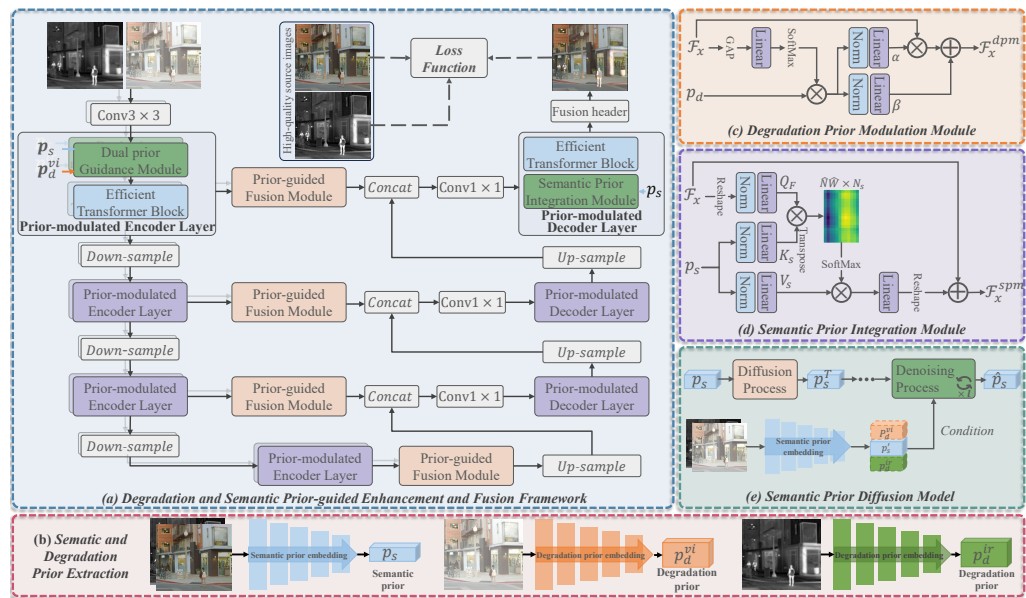

Figure 1: The framework of our degradation and semantic prior dual-guided image fusion network.

However, customizing text for each scenario is costly and impractical. Therefore, it is urgent and challenging to directly identify degradation types from source images, allowing a unified network to effectively handle diverse degradations and achieve optimal information aggregation.

**Unified Image Restoration.** With advancements in deep learning technology, the field of image restoration is evolving beyond designing specialized models for specific degradation factors. Initially, researchers modeled different degradations uniformly and trained task-specific headers (Chen et al., 2021) or separate models (Zamir et al., 2022; Xia et al., 2023) to address various degradations. Furthermore, PromptIR (Potlapalli et al., 2024) and AutoDIR (Jiang et al., 2024) interpret textual user requirements via the CLIP encoder (Radford et al., 2021), guiding the general restoration models to deal with diverse degradations. To avoid reliance on user input, Li et al. (2022) employed contrastive learning to identify degradation types from corrupted images and guide restoration models in addressing corruptions via feature modulation. Similarly, Luo et al. (2024) fine-tuned CLIP on their mixed degradation dataset to develop DA-CLIP, which directly perceives degradation types and predicts high-quality content embeddings from corrupted inputs, aiding restoration networks in handling various degradations. Note that these unified restoration models are designed for natural images and usually not applicable to multi-modal images, such as infrared and visible images.

**Diffusion Model.** Benefiting from their powerful generative capabilities, diffusion models (DMs) have been applied to diverse applications such as text-to-image generation (Rombach et al., 2022), image restoration (Xia et al., 2023), super-resolution (Saharia et al., 2023), deblurring (Chen et al., 2024b), deraining (Özdenizci & Legenstein, 2023), and low-light image enhancement (Yi et al., 2023), consistently delivering impressive results. DMs have also been applied to the image fusion task. Yue et al. (2023) utilized the denoising network of DMs to enhance feature extraction. Zhao et al. (2023b) integrated a pre-trained DM into the EM algorithm, achieving multi-modal fusion with generative priors from natural images. However, these schemes perform the diffusion process in the image domain, making DM-based approaches time-consuming. To improve efficiency, some approaches, such as Stable Diffusion (Rombach et al., 2022), PVDM (Yu et al., 2023), and Hi-Diff (Chen et al., 2024b), transfer the diffusion process into a latent space.

## 3 METHODOLOGY

### 3.1 OVERVIEW

Our workflow is illustrated in Fig. 1. Given low-quality visible image $I_{vi}^{lq}$ and infrared image $I_{ir}^{lq}$, we first extract global low-quality semantic prior ($\hat{p}_s$) and degradation priors ($p_d^{vi}$ and $p_d^{ir}$) via the

semantic prior embedding network (SPEN, $\mathcal{N}_s$) and degradation prior embedding network (DPEN, $\mathcal{N}_d$), described as:

$$\hat{p}_s = \mathcal{N}_s(I_{vi}^{lq}, I_{ir}^{lq}; \theta_s), \qquad \{p_d^{vi}, p_d^{ir}\} = \{\mathcal{N}_d(I_{vi}^{lq}; \theta_d), \mathcal{N}_d(I_{ir}^{lq}; \theta_d)\}. \qquad (1)$$

Then, a semantic prior diffusion model (SPDM, $\mathcal{N}_{dm}$) is designed to restore high-quality semantic prior ($p_s'$) from $\hat{p}_s$ guided by degradation priors, which is formulated as:

$$p_s' = \mathcal{N}_{dm}(\hat{p}_s, p_d^{vi}, p_d^{ir}; \theta_{dm}). \qquad (2)$$

Finally, $p_s'$, $p_d^{vi}$, and $p_d^{ir}$ are employed together to assist the restoration and fusion network ($\mathcal{N}_{ef}$) in synthesizing high-quality fused images ($I_f$):

$$I_f = \mathcal{N}_{ef}(I_{vi}^{lq}, I_{ir}^{lq}, p_d^{vi}, p_d^{ir}, p_s'; \theta_{ef}). \qquad (3)$$

$\mathcal{N}_{ef}$ is a successor of Restormer (Zamir et al., 2022). During feature extraction, we develop two parallel branches to extract multi-scale visible and infrared features, while integrating the semantic and degradation priors to counteract various degradations. The $k$-th level feature extraction is defined as $\mathcal{F}_x^k = E_k(F_x^{k-1}, p_d^x, p_s)$, where $x \in \{ir, vi\}$, $E_k$ denotes the $k$-th level prior-modulated encoder layer. Then, the semantic prior-guided fusion module (PGFM, $\mathcal{M}_f$) is employed to aggregate the complementary information on each level and output $F_f$, formulated as:

$$\mathcal{F}_f^k = \mathcal{M}_f(\mathcal{F}_{ir}^k, \mathcal{F}_{vi}^k, p_s). \qquad (4)$$

A series of prior-modulated decoder layers then refine the fused features from coarse to fine-grained. Finally, a fusion header generates high-quality fusion results ($I_f$). Following previous practice (Xia et al., 2023; Chen et al., 2024b), we train our DSPFusion with a two-stage training strategy, where Stage I focuses on prior extraction and modulation, and Stage II optimizes the SPDM.

## 3.2 STAGE I: PRIOR EXTRACTION AND MODULATION

In Stage I, our purpose is to compress the high- and low-quality images into a compact latent space to characterize scene semantics and degradation types, guiding the restoration and fusion process.

### 3.2.1 NETWORK ARCHITECTURES

**Semantic and Degradation Embedding.** As shown in Fig. 1 (b), high-quality images $I_{vi}^{hq}$ and $I_{ir}^{hq}$ are concatenated and fed into the semantic prior embedding network (SPEN) to obtain a compact semantic prior $p_s$. Similarly, $I_{vi}^{lq}$ and $I_{ir}^{lq}$ are processed separately by the degradation prior embedding network (DPEN) to capture degradation priors $p_d^{vi}$ and $p_d^{ir}$. SPEN and DPEN share a similar structure with residual blocks to generate prior embeddings $p \in \mathbb{R}^{N \times C'}$, where $N$ and $C'$ represent the token number and channel dimension. Notably, $N$ is much smaller than $H \times W$, resulting in a higher compression ratio ($\frac{H \times W}{N_s}$) compared to previous latent diffusion models (*e.g.*, 8) (Rombach et al., 2022), significantly reducing the computational burden of subsequent SPDM. Additionally, the distribution of the latent semantic space ($\mathbb{R}^{N \times C'}$) is simpler than that of the image space ($\mathbb{R}^{H \times W \times 3}$), which can be approximated with fewer iterations. Thus, our SPDM only requires fewer sampling steps ($T \ll 1000$) to infer semantic priors compared to mainstream image-level DM-based fusion schemes (Zhao et al., 2023b), further decreasing computational overhead.

**Dual Prior Guidance Module.** We integrate these priors into $\mathcal{N}_{ef}$ via a dual prior guidance module, consisting of a degradation prior modulation module (DPMM) and a semantic prior integration module (SPIM). Given input features $\mathcal{F}_x$, they first pass through parallel DPMM and SPIM. As shown in Fig. 1 (c), $\mathcal{F}_x$ is compressed into a vector matching the size of $p_d^x$ and then multiplied by $p_d^x$. The resulting product passes through a linear layer to output the modulation parameters $\alpha_d^x$ and $\beta_d^x$. Then, referring to (Li et al., 2022; Yi et al., 2024b), DPMM is formulated as:

$$\mathcal{F}_x^{dpm} = (\alpha_d^x \otimes \mathcal{F}_x) \oplus \beta_d^x. \qquad (5)$$

As a result, DPMM adaptively enhances the features based on the degradation type, enabling various degradations to be addressed with unified model parameters. In parallel, the semantic prior is integrated into $\mathcal{F}_x$ through SPIM to enhance its global perception of high-quality scene context. As shown in Fig. 1 (d), $\mathcal{F}_x$ is mapped as a query $Q_F \in \mathbb{R}^{\hat{H}\hat{W} \times C'}$, and $p_s$ is mapped as the key

$K_s \in \mathbb{R}^{N_s \times C'}$ and value $V_s \in \mathbb{R}^{N_s \times C'}$. Then, cross-attention is applied to perform semantic prior integration and generate the semantic-modulated features as:

$$\mathcal{F}_x^{spi} = \mathcal{F}_x \oplus \text{softmax}\left(Q_F K_s^T / \sqrt{d_k}\right) V_s, \tag{6}$$

where $d_k$ is a learnable scaling factor. Then, we also employ the cross-attention mechanism to aggregate $\mathcal{F}_x^{dpm}$ and $\mathcal{F}_x^{spi}$ to generate final reinforcement features with $\mathcal{F}_x^{dpg} = \mathcal{F}_x^{spi} \oplus \text{softmax}\left(Q_{spi} K_{dpm}^T / \sqrt{d_k}\right) V_{dpm}$, where $Q_{spi}$ is mapped from $\mathcal{F}_x^{spi}$, and $K_{dpm}$ and $V_{dpm}$ are mapped from $\mathcal{F}_x^{dpm}$. Importantly, $p_s$ provides global semantic guidance, and $p_d$ explicitly indicates the degradation types, thereby reducing the overall training difficulty of restoration.

**Prior-guided Fusion Module.** Considering that $p_s$ is jointly extracted from multi-modal inputs, implicitly integrating high-quality and comprehensive scene contexts, we utilize semantic channel attention to generate the channel-wise fusion weight ($w_{ir}^c$ or $w_{vi}^c$) as shown in Fig. 2. Moreover, we employ spatial attention to perform spatial activity level measurements. The infrared (or visible) features are compressed via global max pooling (GMP) and global average pooling (GAP). The pooled results are then concatenated along the channel dimension and fed into a convolutional layer to

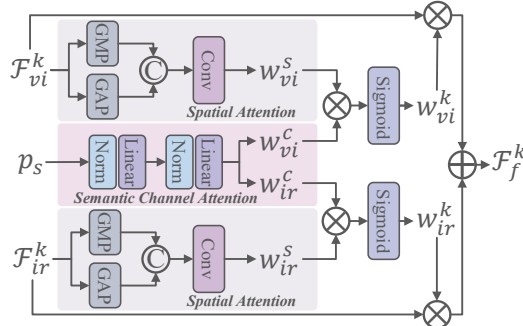

Figure 2: The architecture of the PGFM.

generate spatial weights ($w_{ir}^s$ or $w_{vi}^s$). Subsequently, we comprehensively integrate the channel- and spatial-wise attention to obtain the final fusion weight of the $k$-th layer, formulated as:

$$w_{ir}^k = \sigma(w_{ir}^c \otimes w_{ir}^s), \qquad w_{vi}^k = \sigma(w_{vi}^c \otimes w_{vi}^s), \tag{7}$$

where $\otimes$ denotes element-wise multiplication with broadcasting, and $\sigma$ is the sigmoid function. Finally, the fusion process is defined as: $\mathcal{F}_f^k = w_{ir}^k \mathcal{F}_{ir}^k \oplus w_{vi}^k \mathcal{F}_{vi}^k$.

**Fusion Header.** The multi-scale fused features are refined from coarse to fine using prior-modulated decoder layers, which utilize the semantic prior integration module rather than the dual-prior guidance module, relying exclusively on high-quality scene semantic priors to enhance feature reinforcement. Subsequently, a fusion header, structurally similar to the decoder layer, generates $I_f$ from enhanced fused features ($\mathcal{F}_f^0$). More details can be found in Appendix A.

### 3.2.2 LOSS FUNCTIONS

Since semantic and degradation priors are abstract high-dimensional features without ground-truth constraints, we use the fusion and contrastive losses to jointly optimize $\mathcal{N}_{ef}$, $\mathcal{N}_s$, and $\mathcal{N}_d$. Following Ma et al. (2022) and Yi et al. (2024b), the fusion loss involves the content, structural similarity (SSIM), and color consistency losses. To counteract degradations, we construct these losses with manually obtained high-quality source images. The content loss is defined as:

$$\mathcal{L}_{cont} = \frac{1}{HW}\left(\left\|I_f - \max(I_{vi}^{hq}, I_{ir}^{hq})\right\|_1 + \gamma \cdot \left\|\nabla I_f - \max(\nabla I_{vi}^{hq}, \nabla I_{ir}^{hq})\right\|_1\right), \tag{8}$$

where $\nabla$ denotes the Sobel operator, $\max(\cdot)$ is the maximum selection for preserving salient targets and textures, $\|\cdot\|_1$ and $\gamma$ are the $l_1$-norm and trade-off parameter. The SSIM loss is applied to maintain the structural similarity between the fused image and high-quality sources, formulated as:

$$\mathcal{L}_{ssim} = \left(1 - \text{SSIM}(I_f, I_{vi}^{hq})\right) + \left(1 - \text{SSIM}(I_f, I_{ir}^{hq})\right), \tag{9}$$

where $\text{SSIM}(\cdot, \cdot)$ measures the structural similarity between two images. Referring to Xu et al. (2022a) and Ma et al. (2022), we construct the color consistency loss to encourage fused images to preserve color information from high-quality visible images. It is defined as:

$$\mathcal{L}_{color} = \frac{1}{HW}\left\|\Phi_{CbCr}(I_f) - \Phi_{CbCr}(I_{vi}^{hq})\right\|_1, \tag{10}$$

where $\Phi_{CbCr}(\cdot)$ converts RGB to CbCr. Additionally, our DPEN aims to adaptively identify various degradation types. For inputs with different degradations, the corresponding $p_d$ should be distinct, even if the image contents are the same. To achieve this, we devise a contrastive loss $\mathcal{L}_{cl}$ that pulls together priors characterizing the same degradations while pushing apart priors representing various degradations. For a degradation prior $p_d$, $q_k^+$ and $q_m^-$ are the corresponding positive and negative counterparts. Then, $\mathcal{L}_{cl}$ is formulated as:

$$\mathcal{L}_{cl} = \sum_{k=1}^{K} -\log \frac{\exp(p_d \cdot q_k^+/\tau)}{\sum_m^M \exp(p_d \cdot q_m^-/\tau)}, \tag{11}$$

where $K$ and $M$ denote the number of positive and negative samples, and $\tau$ is a temperature parameter. Specifically, if $p_d$ is extracted from an image with a specific degradation, then $q_k^+$ is extracted from other scenes with the same degradation, while $q_m^-$ is extracted from the same scene but with various degradations or modalities. Finally, the total loss in Stage I for constraining $\mathcal{N}_{ef}$, $\mathcal{N}_s$, and $\mathcal{N}_d$ is the weighted sum of the content, SSIM, color consistency, and contrastive losses:

$$\mathcal{L}_I = \lambda_{cont} \cdot \mathcal{L}_{cont} + \lambda_{ssim} \cdot \mathcal{L}_{ssim} + \lambda_{color} \cdot \mathcal{L}_{color} + \lambda_{cl} \cdot \mathcal{L}_{cl}, \tag{12}$$

where $\lambda_{cont}$, $\lambda_{ssim}$, $\lambda_{color}$, and $\lambda_{cl}$ are hyper-parameters for controlling tradeoff.

### 3.3 STAGE II: SEMANTIC PRIOR DIFFUSION MODEL

In Stage II, we develop a semantic prior diffusion model (SPDM) to restore high-quality semantic prior from low-quality ones, thereby guiding restoration and fusion. Our SPDM builds on conditional denoising diffusion models, involving the forward diffusion and reverse denoising processes, as shown in Fig. 1 (e). In the diffusion process, we first embed $I_{vi}^{hq}$ and $I_{ir}^{hq}$ into a high-quality semantic prior $p_s$, which is simply marked as $x_0$ in this section. $x_0$ serves as the starting point of a forward Markov chain, and gradually adds Gaussian noise to it over $T$ iterations as follows:

$$q(x_{1:T}|x_0) = \prod_{t=1}^{T} q(x_t|x_{t-1}), \qquad q(x_t|x_{t-1}) = \mathcal{N}(x_t; \sqrt{\alpha_t}x_{t-1}, \beta_t\mathbf{I}), \tag{13}$$

where $x_t$ is the $t$-step noisy variable, $\beta_t$ governs the variance of noises, and $\alpha_t = 1 - \beta_t$. Through iterative derivation with reparameterization, the forward Markov process can be reformulated as:

$$q(x_t|x_0) = \mathcal{N}(x_t, \sqrt{\bar{\alpha}_t}x_0, (1 - \bar{\alpha}_t)\mathbf{I}), \tag{14}$$

where $\bar{\alpha}_t = \Pi_{i=1}^{t}\alpha_i$. As $t$ approaches a large value $T$, $\bar{\alpha}_T$ tends to 0 and $q(x_T|x_0)$ approximates the normal distribution $\mathcal{N}(0, \mathbf{I})$, thus completing the forward process.

The reverse process starts from a pure Gaussian distribution and progressively denoises to generate the high-quality semantic prior via a T-step Markov chain, defined as:

$$p(x_{t-1}|x_t) = \mathcal{N}(x_{t-1}; \mu(x_t, t), \sigma_t^2\mathbf{I}), \qquad \mu(x_t, t) = \frac{1}{\sqrt{\alpha_t}}(x_t - \frac{\beta_t}{\sqrt{1 - \bar{\alpha}_t}}\epsilon), \tag{15}$$

where $\sigma_t^2 = \frac{(1-\bar{\alpha}_{t-1})}{(1-\bar{\alpha}_t)}\beta_t$. Following previous works (Ho et al., 2020; Rombach et al., 2022; Chen et al., 2024b), we deploy a denoising U-Net ($\epsilon_\theta$) with the aid of the low-quality semantic prior $\hat{p}_s$ and degradation priors $p_d^{vi}$ and $p_d^{ir}$ to estimate the noise $\epsilon$. Utilizing the reparameterization trick and substituting $\epsilon$ in Eq. (15) with $\epsilon_\theta(x_t, \hat{p}_s, p_d^{vi}, p_d^{ir}, t)$, we can get:

$$x_{t-1} = \frac{1}{\sqrt{\alpha_t}}\left(x_t - \frac{1-\alpha_t}{\sqrt{1-\bar{\alpha}_t}}\epsilon_\theta(x_t, \hat{p}_s, p_d^{vi}, p_d^{ir}, t)\right) + \sigma_t z. \tag{16}$$

Traditionally, the objective for training $\epsilon_\theta$ is defined as the weighted variational bound:

$$\nabla_\theta \|\epsilon_t - \epsilon_\theta(\sqrt{\bar{\alpha}_t}x_0 + \sqrt{1 - \bar{\alpha}_t}\epsilon_t, \hat{p}_s, p_d^{vi}, p_d^{ir}, t)\|_2^2. \tag{17}$$

Since the distribution of the latent semantic space ($\mathbb{R}^{N_s \times C'}$) is simpler than that of the image space ($\mathbb{R}^{H \times W \times 3}$), the semantic prior ($p_s'$) can be generated with fewer iterations (Chen et al., 2024b). Thus, we run complete $T$ ($\ll 1000$) iterations of the reverse process to infer $p_s'$. Consequently, we use $\mathcal{L}_{diff} = \|p_s' - p_s\|_1$ to train SPDM. We also apply content, SSIM, and color consistency losses to collaboratively constrain the training of SPDM. Thus, the total loss of Stage II is defined as:

$$\mathcal{L}_{II} = \lambda_{diff} \cdot \mathcal{L}_{diff} + \lambda_{cont} \cdot \mathcal{L}_{cont} + \lambda_{ssim} \cdot \mathcal{L}_{ssim} + \lambda_{color} \cdot \mathcal{L}_{color}, \tag{18}$$

where $\lambda_{diff}$ is a hyper-parameter for balancing various losses.

Table 1: Quantitative comparison results on typical fusion datasets. The best and second-best results are highlighted in Red and Blue, respectively.

| Methods | MSRS | | | | LLVIP | | | | RoadScene | | | | TNO | | | |
|---|---|---|---|---|---|---|---|---|---|---|---|---|---|---|---|---|
| | EN | MI | VIF | Qabf | EN | MI | VIF | Qabf | EN | MI | VIF | Qabf | EN | MI | VIF | Qabf |
| DeFus. | 6.350 | 3.054 | 0.736 | 0.505 | 7.112 | 3.196 | 0.683 | 0.487 | 6.910 | 3.018 | 0.537 | 0.404 | 6.581 | 2.917 | 0.596 | 0.384 |
| PAIF | 5.830 | 2.907 | 0.470 | 0.329 | 6.937 | 2.533 | 0.432 | 0.286 | 6.750 | 2.919 | 0.387 | 0.248 | 6.198 | 2.561 | 0.421 | 0.243 |
| MetaFus. | 6.355 | 1.693 | 0.700 | 0.476 | 6.645 | 1.400 | 0.629 | 0.429 | 7.363 | 2.195 | 0.517 | 0.416 | 7.184 | 1.815 | 0.615 | 0.362 |
| LRRNet | 6.197 | 2.886 | 0.536 | 0.451 | 6.006 | 1.749 | 0.397 | 0.281 | 7.051 | 2.649 | 0.463 | 0.344 | 6.944 | 2.577 | 0.577 | 0.352 |
| MURF | 5.036 | 1.516 | 0.403 | 0.311 | 5.869 | 2.017 | 0.355 | 0.317 | 6.961 | 2.492 | 0.498 | 0.468 | 6.654 | 1.912 | 0.528 | 0.378 |
| SegMiF | 6.109 | 2.472 | 0.774 | 0.565 | 7.172 | 2.819 | 0.837 | 0.651 | 7.254 | 2.657 | 0.615 | 0.543 | 6.976 | 3.036 | 0.876 | 0.589 |
| DDFM | 6.182 | 2.661 | 0.721 | 0.468 | 6.814 | 2.590 | 0.632 | 0.475 | 7.111 | 2.84 | 0.587 | 0.482 | 6.878 | 2.408 | 0.691 | 0.466 |
| EMMA | 6.713 | 4.129 | 0.957 | 0.632 | 7.160 | 3.374 | 0.740 | 0.461 | 7.383 | 3.140 | 0.605 | 0.461 | 7.203 | 3.038 | 0.755 | 0.472 |
| Text-IF | 6.648 | 4.283 | 1.031 | 0.692 | 6.961 | 3.142 | 0.855 | 0.648 | 7.299 | 2.988 | 0.698 | 0.588 | 7.168 | 3.524 | 0.918 | 0.583 |
| DSPFusion | 6.695 | 4.736 | 1.044 | 0.726 | 7.314 | 4.390 | 0.943 | 0.717 | 7.363 | 3.962 | 0.755 | 0.667 | 7.152 | 4.680 | 0.931 | 0.640 |

## 4 EXPERIMENTS

### 4.1 EXPERIMENTAL DETAILS

**Implementation Details.** Our restoration and fusion network inherits Restormer (Zamir et al., 2022), which is a 4-level encoder-decoder Transformer architecture with degradation and semantic prior modulation. From level-1 to level-4, the numbers of Transformer blocks are set as $[2, 2, 4, 4]$, and the channel number is set as $[32, 64, 128, 256]$. The SPEN contains 6 residual blocks, whose token number and channel dimension are set to $N_c = 16$ and $C'_c = 256$. The DPEN contains 4 residual blocks, whose token number and channel dimension are set to $N_d = 16$ and $C'_d = 128$. The time-step of SPDM is set as $T = 10$. We train our DSPFusion with the AdamW optimizer with $\beta_1 = 0.9$ and $\beta_2 = 0.99$. The learning rate is initialized to $2 \times 10^{-4}$ and gradually reduced to $1 \times 10^{-6}$ with cosine annealing. In both Stages I and II, the training involves $50k$ iterations. In the initial $30k$ iterations, the patch and batch sizes are set to 224 and 4, and in subsequent $20k$ iterations, the patch and batch sizes are set to 256 and 3. The hyper-parameters are empirically set as $\gamma = 0.75$, $\lambda_{cont} = 15, \lambda_{ssim} = 2, \lambda_{color} = 20, \lambda_{cl} = 1, \lambda_{diff} = 10$. The numbers of positive and negative samples are set to $K = 3$ and $M = 7$. Our training data is construed on the EMS dataset (Yi et al., 2024b), including $2, 210$ scenarios, with $8, 804$ and $10, 318$ low-quality infrared and visible images.

**Experiment Configurations.** We first demonstrate the fusion performance on four typical datasets, *i.e.*, MSRS (Tang et al., 2022c), LLVIP (Jia et al., 2021), RoadScene (Xu et al., 2022a), TNO (Toet, 2017), with four quantitative metrics, including EN, MI, VIF, and $Q_{abf}$. The numbers of test images in the MSRS, LLVIP, RoadScene, and TNO datasets are 361, 50, 50, and 25, respectively. We compare our DSPFusion with nine SOTA fusion methods, including DeFusion (Liang et al., 2022), PAIF (Liu et al., 2023b), MetaFusion (Zhao et al., 2023a), LRRNet (Li et al., 2023a), MURF (Xu et al., 2023), SegMiF (Liu et al., 2023a), DDFM (Zhao et al., 2023b), EMMA (Zhao et al., 2024), Text-IF (Yi et al., 2024b). We validate the performance of DSPFusion under various degradations, including blur, rain, low-light, over-exposure, and random noise in visible images (VI), as well as low-contrast, random noise, and stripe noise in infrared images (IR). We also evaluate the robustness of DSPFusion under mixed degradations, *i.e.*, rain or low-light in VI, and low-contrast or stripe noise in IR. All scenarios include 100 test samples, except for the over-exposed scenario in visible images, which contains 50 test samples. Four no-reference metrics, *i.e.*, MUSIQ, PI, TReS, and SD (or EN or SF), are utilized to evaluate the quality of the fused images. Some SOTA image enhancement algorithms are deployed to pre-enhance low-quality sources for fair comparisons. In particular, Hi-Diff (Chen et al., 2024b) for deblurring, NeRD-Rain (Chen et al., 2024a) for deraining, Spadap (Li et al., 2023b) for denoising, QuadPrior (Wang et al., 2024) for low-light enhancement, IAT (Cui et al., 2022) for exposure correction, WDNN (Guan et al., 2019) for stripe noise removal, and the method in Tang et al. (2022c) for low-contrast enhancement.

### 4.2 FUSION PERFORMANCE COMPARISON

**Comparison without Pre-enhancement.** Table 1 shows quantitative results on typical fusion datasets. DSPFusion achieves superior performance in MI and Qabf, effectively transferring complementary and edge information into fused images. The optimal VIF indicates that our fused images exhibit excellent visual perception quality, while the comparable EN suggests that our fusion results

Table 2: Quantitative comparison results in degraded scenarios with enhancement.

| Methods | VI (Blur) | | | | VI (Rain) | | | | VI (Low-light, LL) | | | | VI (Over-exposure, OE) | | | |
|---|---|---|---|---|---|---|---|---|---|---|---|---|---|---|---|---|
| | MUSIQ | PI | TReS | SF | MUSIQ | PI | TReS | SD | MUSIQ | PI | TReS | SD | MUSIQ | PI | TReS | SD |
| DeFus. | 38.971 | 4.368 | 36.968 | 8.765 | 44.182 | 3.575 | 44.516 | 40.471 | 43.645 | 3.510 | 42.961 | 36.326 | 46.316 | 3.367 | 45.972 | 38.649 |
| PAIF | 39.363 | 5.431 | 40.298 | 9.102 | 45.685 | 4.720 | 46.819 | 38.593 | 42.432 | 4.725 | 45.248 | 39.239 | 37.032 | 5.261 | 35.094 | 52.062 |
| MetaFus. | 36.674 | 4.946 | 34.451 | 18.167 | 41.160 | 4.127 | 39.105 | 49.808 | 38.775 | 4.061 | 33.915 | 46.852 | 43.590 | 3.217 | 38.889 | 52.473 |
| LRRNet | 41.490 | 4.074 | 41.774 | 11.084 | 48.591 | 3.218 | 50.646 | 42.612 | 44.037 | 3.391 | 47.584 | 31.204 | 48.263 | 2.803 | 51.459 | 44.852 |
| MURF | 45.860 | 3.379 | 46.603 | 12.374 | 48.703 | 3.131 | 49.965 | 25.117 | 44.979 | 3.144 | 46.383 | 21.418 | 52.484 | 2.518 | 56.603 | 33.128 |
| SegMiF | 42.907 | 4.02 | 41.349 | 12.636 | 47.850 | 2.909 | 50.016 | 46.861 | 45.16 | 3.049 | 46.311 | 45.614 | 51.282 | 2.688 | 54.419 | 50.445 |
| DDFM | 42.005 | 3.928 | 42.765 | 9.474 | 47.117 | 3.229 | 49.757 | 56.140 | 44.022 | 3.346 | 47.090 | 32.197 | 52.182 | 2.435 | 59.047 | 41.834 |
| EMMA | 41.442 | 4.128 | 37.81 | 14.036 | 48.315 | 3.078 | 45.750 | 55.989 | 45.124 | 3.201 | 42.383 | 45.881 | 49.446 | 2.651 | 48.389 | 54.538 |
| Text-IF | 44.536 | 3.665 | 47.524 | 15.153 | 50.109 | 2.775 | 56.966 | 54.842 | 46.015 | 2.994 | 50.279 | 51.537 | 52.048 | 2.290 | 56.979 | 52.200 |
| DSPFusion | 47.137 | 2.972 | 49.750 | 15.693 | 50.467 | 2.557 | 56.528 | 55.599 | 48.500 | 2.768 | 54.090 | 45.940 | 52.812 | 2.206 | 57.198 | 54.840 |

| Methods | VI (Random noise, RN) | | | | IR (Low-contrast, LC) | | | | IR (Random noise, RN) | | | | IR (Stripe noise, SN) | | | |
|---|---|---|---|---|---|---|---|---|---|---|---|---|---|---|---|---|
| | MUSIQ | PI | TReS | EN | MUSIQ | PI | TReS | SD | MUSIQ | PI | TReS | EN | MUSIQ | PI | TReS | EN |
| DeFus. | 34.661 | 4.883 | 30.837 | 7.028 | 44.378 | 3.538 | 45.154 | 39.753 | 40.463 | 3.735 | 43.565 | 7.065 | 42.603 | 3.593 | 42.966 | 6.968 |
| PAIF | 33.875 | 6.306 | 33.362 | 6.731 | 47.192 | 4.525 | 47.946 | 38.499 | 47.198 | 4.456 | 49.016 | 6.772 | 47.403 | 4.482 | 48.921 | 6.747 |
| MetaFus. | 32.918 | 4.969 | 27.623 | 7.362 | 40.136 | 4.224 | 39.847 | 55.705 | 40.236 | 4.304 | 39.461 | 7.419 | 39.543 | 4.215 | 38.837 | 7.448 |
| LRRNet | 34.856 | 4.642 | 32.07 | 7.108 | 48.540 | 3.189 | 50.456 | 42.697 | 46.625 | 3.327 | 49.883 | 7.015 | 47.382 | 3.248 | 48.708 | 7.016 |
| MURF | 40.866 | 3.386 | 42.447 | 6.233 | 48.958 | 3.063 | 51.079 | 26.875 | 49.799 | 3.276 | 49.431 | 6.204 | 47.135 | 3.202 | 47.908 | 6.223 |
| SegMiF | 37.362 | 4.081 | 31.640 | 6.975 | 48.335 | 2.803 | 51.976 | 50.813 | 47.909 | 2.977 | 50.489 | 6.997 | 46.971 | 2.852 | 49.394 | 7.080 |
| DDFM | 37.325 | 4.351 | 36.870 | 6.940 | 48.039 | 3.222 | 51.181 | 37.090 | 46.829 | 3.512 | 48.653 | 6.960 | 45.310 | 3.361 | 47.234 | 6.908 |
| EMMA | 34.754 | 4.432 | 29.364 | 7.44 | 48.721 | 2.978 | 46.842 | 58.299 | 45.870 | 3.124 | 45.667 | 7.453 | 47.286 | 3.036 | 45.717 | 7.439 |
| Text-IF | 39.200 | 3.930 | 36.588 | 7.406 | 50.022 | 2.794 | 55.203 | 56.117 | 48.795 | 2.944 | 54.023 | 7.402 | 49.376 | 2.829 | 53.717 | 7.440 |
| DSPFusion | 47.718 | 2.954 | 52.787 | 7.343 | 50.597 | 2.623 | 56.988 | 56.060 | 50.752 | 2.852 | 57.266 | 7.349 | 51.055 | 2.787 | 57.243 | 7.353 |

| Methods | VI (Rain) and IR (LC) | | | | VI (Rain) and IR (SN) | | | | VI (LL) and IR (LC) | | | | VI (LL) and IR (SN) | | | |
|---|---|---|---|---|---|---|---|---|---|---|---|---|---|---|---|---|
| | MUSIQ | PI | TReS | SD | MUSIQ | PI | TReS | EN | MUSIQ | PI | TReS | SD | MUSIQ | PI | TReS | EN |
| DeFus. | 44.168 | 3.727 | 44.601 | 35.803 | 41.933 | 3.808 | 42.020 | 6.827 | 42.000 | 3.654 | 41.607 | 30.416 | 40.279 | 3.684 | 38.722 | 6.788 |
| PAIF | 46.172 | 4.817 | 46.361 | 37.658 | 46.013 | 4.778 | 47.135 | 6.661 | 42.671 | 4.639 | 46.503 | 30.425 | 41.686 | 4.796 | 44.084 | 6.616 |
| MetaFus. | 40.528 | 4.221 | 39.237 | 51.363 | 40.175 | 4.214 | 38.272 | 7.339 | 38.832 | 3.833 | 33.506 | 43.002 | 37.957 | 4.086 | 31.719 | 7.194 |
| LRRNet | 48.396 | 3.248 | 50.05 | 42.434 | 47.151 | 3.335 | 48.290 | 6.999 | 43.231 | 3.414 | 45.943 | 31.170 | 42.730 | 3.564 | 44.534 | 6.630 |
| MURF | 49.221 | 3.082 | 50.669 | 26.013 | 47.048 | 3.271 | 47.253 | 6.148 | 44.943 | 3.117 | 45.792 | 18.915 | 43.006 | 3.300 | 42.568 | 5.910 |
| SegMiF | 48.392 | 2.915 | 50.802 | 46.525 | 46.548 | 2.998 | 47.923 | 6.989 | 44.512 | 2.965 | 45.744 | 40.511 | 43.337 | 3.218 | 43.201 | 6.982 |
| DDFM | 47.513 | 3.318 | 50.074 | 36.381 | 44.869 | 3.447 | 46.338 | 6.873 | 43.242 | 3.473 | 46.691 | 29.616 | 41.600 | 3.788 | 43.711 | 6.722 |
| EMMA | 48.864 | 3.017 | 46.307 | 54.659 | 47.216 | 3.119 | 45.023 | 7.366 | 44.251 | 3.211 | 41.010 | 43.038 | 43.652 | 3.233 | 39.788 | 7.209 |
| Text-IF | 50.380 | 2.765 | 56.429 | 56.381 | 48.221 | 2.885 | 53.446 | 7.400 | 47.815 | 2.915 | 49.334 | 45.733 | 39.866 | 3.272 | 39.043 | 7.095 |
| DSPFusion | 50.672 | 2.607 | 56.362 | 55.686 | 50.883 | 2.737 | 57.098 | 7.349 | 48.547 | 2.819 | 53.892 | 46.144 | 48.711 | 3.020 | 54.598 | 7.263 |

Figure 3: Visualization of fusion results in degraded scenarios with enhancement.

retain abundant information. In summary, the quantitative results demonstrate the remarkable fusion performance of our method. Some visual fusion results are provided in Appendix C.

**Comparison with Pre-enhancement.** It is worth mentioning that almost all fusion algorithms apply state-of-the-art image restoration methods to pre-enhance source images for fair comparisons. Notably, Text-IF utilizes its built-in enhancement module for low-light and overexposed visible images, as well as low-contrast and random noise in infrared images, while applying pre-processing algorithms for other degraded scenarios. Moreover, when source images are affected by random noise, PAIF does not employ additional denoising algorithms for pre-enhancement, as its fusion network is inherently robust to noise. The quantitative results in degraded scenarios are shown in

Table 3: Quantitative results of detection.

| Methods | Fusion in nighttime scenarios with enhancement | | | | | |
|---|---|---|---|---|---|---|
| | Prec. | Recall | AP@50 | AP@75 | AP@95 | mAP |
| DeFus. | 0.983 | 0.831 | 0.911 | 0.802 | 0.057 | 0.684 |
| PAIF | **0.989** | 0.848 | 0.919 | 0.799 | 0.071 | 0.683 |
| MetaFus. | 0.958 | 0.871 | 0.927 | 0.806 | 0.116 | 0.697 |
| LRRNet | **0.989** | 0.811 | 0.902 | 0.783 | 0.040 | 0.668 |
| MURF | 0.980 | **0.884** | 0.935 | 0.809 | 0.095 | 0.707 |
| SegMiF | 0.981 | 0.835 | 0.910 | 0.799 | 0.092 | 0.687 |
| DDFM | 0.983 | 0.842 | 0.917 | 0.801 | 0.067 | 0.679 |
| EMMA | 0.971 | 0.846 | 0.916 | 0.791 | 0.085 | 0.685 |
| Text-IF | **0.989** | 0.782 | 0.887 | 0.778 | 0.043 | 0.667 |
| Ours | 0.974 | **0.884** | **0.936** | **0.822** | **0.169** | **0.726** |

Table 4: Results of computational efficiency.

| Methods | Fusion | | | Fusion with enhancement | | |
|---|---|---|---|---|---|---|
| | parm.(m) | flops(g) | time(s) | parm.(m) | flops(g) | time(s) |
| DeFus. | 7.874 | 71.55 | 0.075 | 234.81 | 869.24 | 0.478 |
| PAIF | 44.86 | 122.12 | 0.052 | 271.80 | 919.80 | 0.455 |
| MetaFus. | 0.812 | 159.48 | **0.028** | 227.74 | 957.16 | 0.431 |
| LRRNet | **0.049** | **14.17** | 0.085 | 226.98 | 811.86 | 0.488 |
| MURF | 0.120 | 31.50 | 0.205 | 227.05 | 829.19 | 0.608 |
| SegMiF | 45.04 | 353.7 | 0.147 | 271.97 | 1151.4 | 0.550 |
| DDFM | 552.7 | 5220. | 34.50 | 779.59 | 6018.2 | 34.91 |
| EMMA | 1.516 | 41.54 | 0.037 | 228.45 | 839.23 | 0.440 |
| Text-IF | 89.01 | 1518.9 | 0.157 | 89.01 | 1518.9 | 0.157 |
| Ours | 13.99 | 254.34 | 0.119 | **13.99** | **254.34** | **0.119** |

Table 2. Our DSPFusion achieves the best MUSIQ, PI, and TReS in almost all degraded scenarios, demonstrating its effectiveness in mitigating degradation, aggregating complementary context, and producing high-quality fused images. We apply various no-reference statistical metrics, *i.e.*, SD, EN, or SF, to evaluate fusion results on different degraded scenarios according to the properties of degradations. DSPFusion exhibits comparable performance to other methods on these metrics.

Qualitative comparison results are presented in Fig. 3. When source images are affected by noise, denoising algorithms can remove noise but often at the cost of blurring fine details, such as the text on the wall and the license plate. By contrast, DSPFusion preserves rich texture while suppressing noise. Moreover, in blurry scenarios, although HI-Diff can partially mitigate the blur, DSPFusion delivers sharper visual clarity. This advantage arises from the fact that *our method can enhance the degraded modality by leveraging comprehensive semantic priors from both infrared and visible sources, offering a more complete scene representation*. Conversely, single-modality enhancement approaches rely solely on limited intra-modality information to infer degradation-free images, which naturally limits their enhancement performance. As shown in Fig. 3 (e) and (f), our method can effectively handle challenging scenarios where both infrared and visible images suffer from degradations. This is achieved by *employing modality-specific degradation priors in a divide-and-conquer manner to modulate the features of each modality individually, ensuring that the feature enhancement is precisely adapted to the unique characteristics of each modality*. Both quantitative and qualitative results demonstrate the superiority of our DSPFusion in suppressing degradations and integrating complementary information across various degraded scenarios within a unified model.

### 4.3 EXTENDED EXPERIMENTS AND DISCUSSIONS

**Degradation Prior Visualization.** Figure 4 shows t-SNE visualizations illustrating the ability of different models to distinguish degradation types. While DA-CLIP can partially separate degradations in visible images, it performs poorly on infrared images.

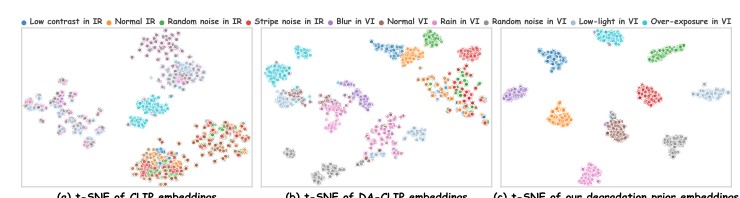

Figure 4: t-SNE plots of degradation types.

In contrast, our DPEN effectively distinguishes degradations across modalities, laying a solid foundation for subsequent information restoration and fusion.

**Object Detection.** We also evaluate object detection performance on LLVIP to indirectly assess the fusion quality using retrained YOLOv8 (Redmon et al., 2016). Qualitative and quantitative results are shown in Fig. 5 and Tab. 3. Owing to superior informa-

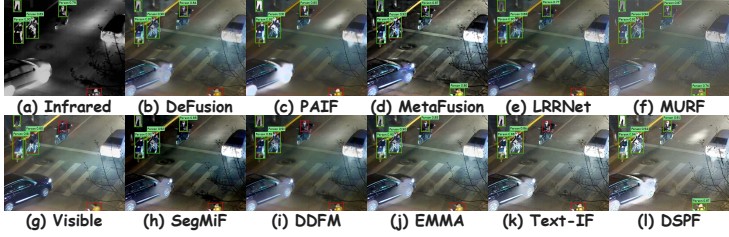

Figure 5: Visual comparison of object detection.

tion restoration and integration, the detector identifies all pedestrians in our fusion results with higher confidence and achieves the best average precision (AP) across various confidence thresholds.

**Evaluation with DepictQA.** We introduce DepictQA (You et al., 2024), a descriptive image quality assessment metric based on the vision language models, to evaluate our fused image quality. As shown in Fig. 6, the infrared image suffers from significant

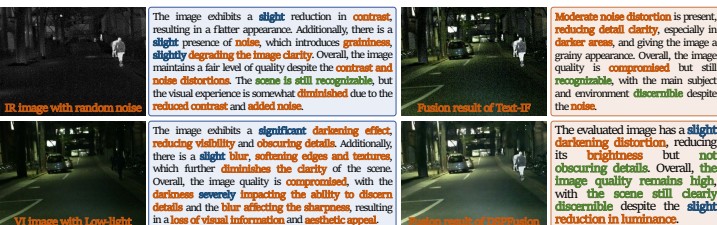

Figure 6: Evaluation results with DepictQA.

noise, while the visible image is affected by low-light. DepictQA not only accurately identifies these degradations but also describes their severity. Although Text-IF only mildly suppresses degradations, thanks to effective information aggregation, DepictQA judges that while its fusion result experiences moderate noise distortion, it remains recognizable. In contrast, DSPFusion successfully achieves low-light enhancement and noise reduction, along with effective information aggregation. Therefore, DepictQA assesses our image quality as remaining high.

**Computational Efficiency.** We conduct the diffusion process in a compact space, greatly reducing computational costs. As shown in Tab. 4, compared to DDFM, which performs diffusion in the image space, DSPFusion exhibits a significant advantage in computational efficiency, being over $200\times$ faster than DDFM. Moreover, in comparison to Text-IF, which relies on an additional CLIP model for degradation prompting, DSPFusion also offers a notable improvement in efficiency. Specifically, in degraded scenarios, it offers a clear advantage by obviating the need for additional pre-processing.

**Discussion on Compound Degradations.** As mentioned above, our DSPFusion can effectively handle scenarios with a single degradation type across multiple modalities in a unified model. However, as shown in Fig. 7, when one

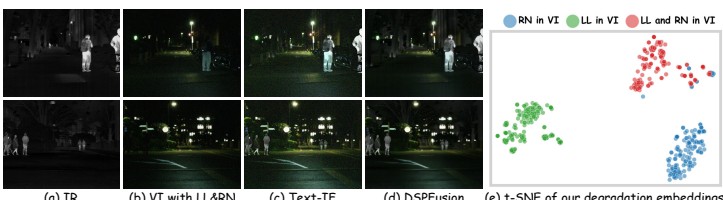

Figure 7: A schematic of the failure cases.

modality experiences compound degradations, it only addresses the dominant degradation, despite DPEN encoding degradation priors into a distinct feature space. Notably, although we prompt Text-IF that the visible image suffers from both noise and low-light degradations, it still struggles to resolve these issues because the coupled text embeddings are unfamiliar to the model.

**Ablation Studies.** In order to demonstrate the effectiveness of our specific designs, we conduct ablation studies by individually removing either DPEN or

Table 5: Quantitative results of ablation study.

| | Deg. prior | Sema. prior | VI (Low-light, LL) | | | | IR (Low-contrast, LC) | | | | VI (LL) and IR (LC) | | | |
|---|---|---|---|---|---|---|---|---|---|---|---|---|---|---|
| | | | MUSIQ | PI | Tres | SD | MUSIQ | PI | Tres | SD | MUSIQ | PI | Tres | SD |
| I | ✗ | ✗ | 47.71 | 2.83 | 52.34 | **46.41** | 50.35 | 2.74 | 56.17 | 55.31 | 47.67 | 2.89 | 52.24 | 46.05 |
| II | ✓ | ✗ | 48.39 | 2.78 | 53.51 | 45.13 | 50.53 | 2.68 | 56.71 | 55.96 | 48.35 | 2.84 | 52.98 | 45.16 |
| III | ✗ | ✓ | 48.26 | **2.75** | 53.87 | 45.24 | 50.29 | **2.50** | **57.03** | 55.15 | 47.77 | **2.61** | 53.52 | 42.08 |
| Ours | ✓ | ✓ | **48.50** | 2.77 | **54.09** | 45.94 | **50.60** | 2.62 | 56.99 | **56.06** | **48.55** | 2.82 | **53.89** | 46.14 |

SPEN in various degraded scenarios, including low-light in visible images, low-contrast in infrared images, and their combination. From Tab. 5, one can find that both DPEN and SPEN play crucial roles in improving the performance of DSPFusion. Particularly, our method better reconciles degradation suppression with information aggregation by integrating degradation and semantic priors.

## 5 CONCLUSION

This work presents a degradation and semantic prior dual-guided framework for degraded image fusion. A degradation prior embedding network is designed to extract modality-specific degradation priors, guiding the unified model to purposefully address degradations. A semantic prior embedding network is developed to capture semantic prior from cascaded source images, enabling implicit complementary information aggregation. Moreover, we devise a semantic prior diffusion model to restore high-quality scene priors in a compact space, providing global semantic guidance for subsequent restoration and fusion. Experiments on multiple degraded scenarios demonstrate the superiority of our method in suppressing degradation and aggregating information.

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

## A    MORE DETAILS ABOUT METHODOLOGY DESIGNS

In this section, we provide more details and interpretations about our methodology designs. As illustrated in Fig. 1 (a), unlike the prior-modulated encoder layer, the prior-modulated decoder layer employs the semantic prior integration module instead of the dual-prior guidance module, relying solely on high-quality scene semantic priors to support feature reinforcement. This design aims to effectively eliminate the influence of degradation factors during the feature encoding stage with the assistance of degradation and semantic priors. After feature fusion, if the fused features still contain mixed degradations, their distribution will differ from that of a single degradation, making it challenging for the degradation prior to accurately characterize them.

Figure 8 presents the schematic diagram of our contrastive mechanism for constraining the degradation prior embedding network. The basic construction process of the contrastive loss is outlined in Section 3.2.2. This section focuses on the criteria for selecting the positive and negative samples. The number of positive samples, $K$, is set to 3, and the number of negative samples, $M$, is set to 7.

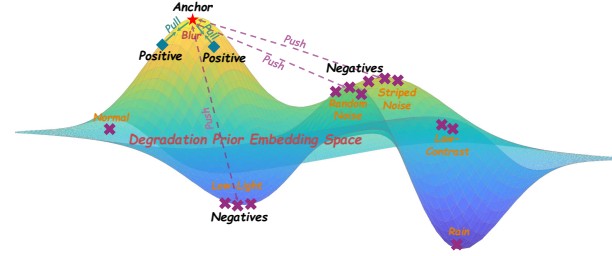

Figure 8: Schematic diagram of the contrastive mechanism.

For instance, the degradation types of the anchors in visible and infrared images are low-light and low-contrast, respectively. For visible images, the positive samples consist of 3 low-light visible images from different scenes, while for infrared images, the positive samples are 3 low-contrast infrared images from different scenes. We then select 6 negative samples for both visible and infrared anchors from the remaining visible or infrared images with the same scene content as the anchors, where the visible images suffer from over-exposure, blur, rain, random noise, or no degradation, while the infrared images are affected by random noise, stripe noise, or no degradation. Moreover, the infrared anchor is added as the negative sample for the visible anchor, and vice versa. Therefore, for each anchor, there are 3 positive samples and 7 negative samples.

## B    MORE EXPERIMENT DETAILS

### B.1    IMPLEMENTATION DETAILS

We construct the training data on the EMS dataset [1], where the degradation types for visible images include blur, rain, low-light, over-exposure, and random noise, and the degradations for infrared images include low-contrast, random noise, and stripe noise. We further extend this dataset by introducing low-light scenes from the MSRS dataset, where the visible images are enhanced by Quad-Prior (Wang et al., 2024). Finally, our training dataset consists of $2,210$ paired high-quality infrared and visible images. The source infrared images include $2,210$ degradation-free images, $2,210$ low-contrast images, $2,210$ images with random noise, and $2,210$ images with stripe noise. The source visible images involve $2,210$ degradation-free images, $2,210$ blurred images, $2,210$ rain-affected images, $2,210$ images with random noise, $1,316$ low-light images, and $136$ over-exposed images.

### B.2    EXPERIMENT CONFIGURES

In the degraded scenarios, we first use no-reference image quality assessment metrics, *i.e.,* MUSIQ, PI, and TReS, with a lower value indicating better performance for the PI metric. We also utilize statistical metrics frequently employed in the image fusion field to assess performance based on the properties of degradations. In detail, when source images are affected by blurring, textures become obscured. Therefore, we use the SF metric to evaluate the richness of details in the fusion results. Additionally, when source images suffer from issues such as low light, overexposure, or low contrast, the overall contrast diminishes. Consequently, we use the SD metric to assess the effectiveness of the fusion results in counteracting these degradations. Furthermore, when images are affected by noise or rain, both SF and SD values may be artificially inflated, so we employ the EN metric to

---

[1]https://github.com/XunpengYi/EMS

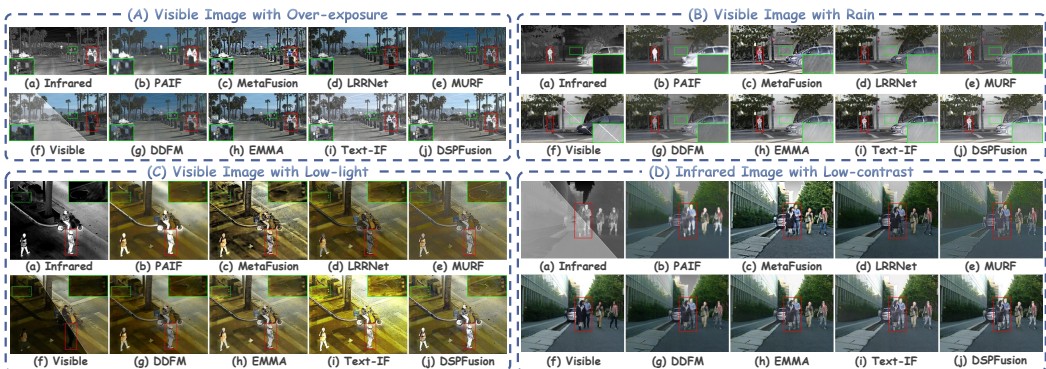

Figure 9: Visualization of fusion results on the typical fusion datasets.

Figure 10: Visualization of fusion results in degraded scenarios with pre-enhancement.

evaluate the fusion performance accurately. All experiments are conducted on the NVIDIA RTX 4090 GPUs and 2.50 GHz Intel(R) Xeon(R) Platinum 8180 CPUs with PyTorch.

## C MORE RESULTS AND ANALYSIS

Figure 9 presents representative visual fusion results on the MSRS and LLVIP datasets. We can find that MetaFusion, LRRNet, MURF, and DDFM diminish the prominence of thermal targets, while PAIF, EMMA, and Text-IF struggle to outline streetlights and headlights in overexposed conditions. In contrast, DSPFusion simultaneously highlights significant targets and preserves abundant textures. Overall, the quantitative and qualitative results in Tab. 1 and Fig. 9 collectively demonstrate the impressive fusion performance of our DSPFusion.

Figure 10 provides more fusion results in the degraded scenarios with enhancement. From Figs. 3 and 10, one can find that PAIF obscures texture details within the scenes, particularly in prominent infrared targets, despite the excessive enhancement of these targets. This is attributed to PAIF attempting to counteract noise. Additionally, MetaFusion introduces artificial textures during the fusion process, which is the primary factor for its higher SF metric. We believe this is caused by MetaFusion paying more attention to the object detection task, resulting in insufficient consideration for visual perception. LRRNet, MURF, and DDFM seem to simply neutralize infrared and visible images, resulting in their fusion results that reduce the prominence of infrared targets and cause a loss of texture details in the visible images. EMMA relies on manually selected fused images from existing fusion algorithms for supervision, which limits its performance potential. For instance, while EMMA can aggregate complementary information from source images across most scenarios, it may slightly diminish the prominence of infrared targets. Although Text-IF demonstrates good fusion performance, it still has several notable shortcomings. Firstly, Text-IF is highly sensitive to text prompts. As shown in Fig. 3 (f), when we prompt it that the visible and infrared images suffer from degradations (such as low-light and low-contrast) simultaneously, it fails to mitigate the effects of degradations, even though it handles individual degradations effectively, as demonstrated in Fig. 10 (C) and (D). This may be caused by the fact that the feature embedding of such coupled text prompts is unfamiliar to the pre-trained model. Moreover, it is limited to addressing only a few specific types of degradations, such as low-light and over-exposure in visible images as well as random noise and low-contrast in infrared images. In contrast, our method adaptively identifies degradation types from source images, enabling it to effectively handle the common degradations and achieve

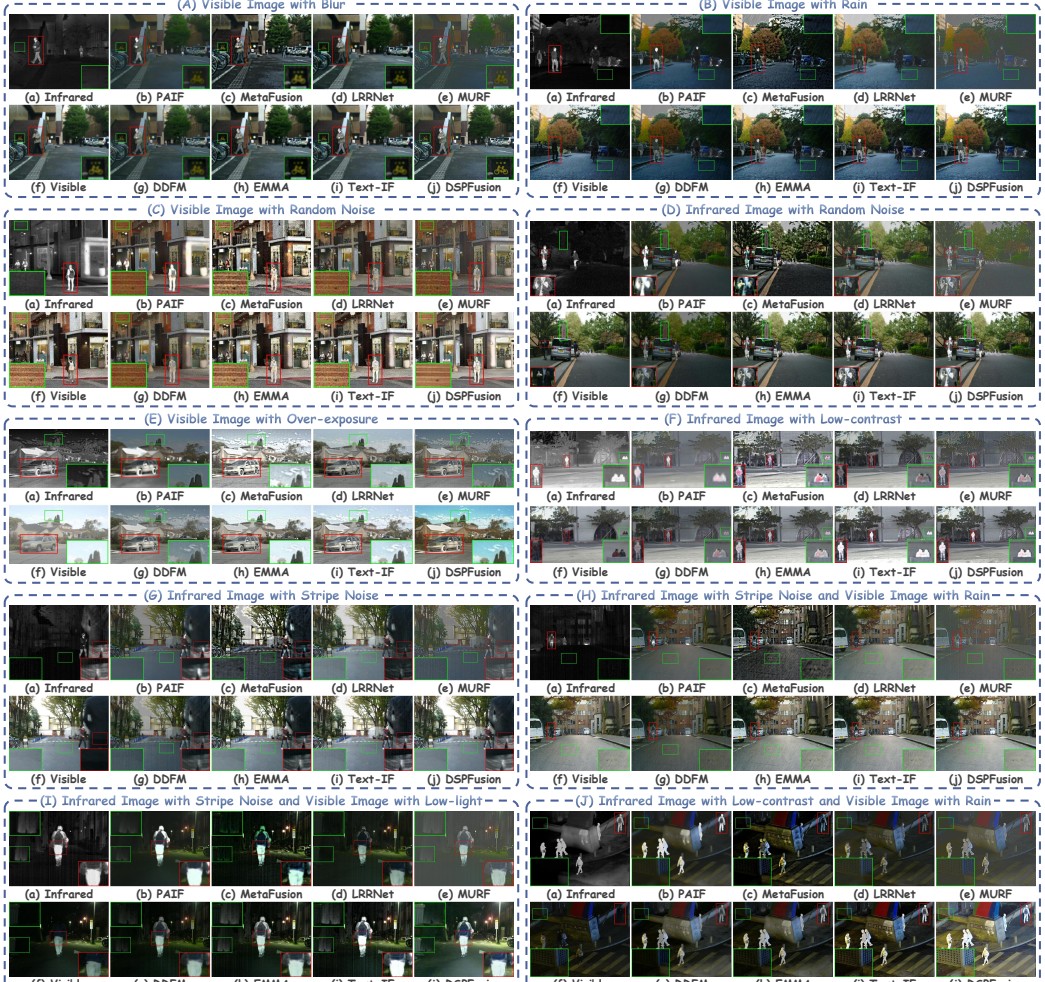

Figure 11: Visualization of fusion results in degraded scenarios without pre-enhancement.

complementary information aggregation within a unified model. Moreover, by employing a divide-and-conquer manner to address degradations in infrared and visible images separately, it remains effective even when both infrared and visible images suffer from degradations simultaneously.

Figure 11 presents the qualitative comparison results in degraded scenarios without pre-enhancement. It is evident that although most fusion algorithms can effectively aggregate complementary information, they are hindered by degradations and cannot provide satisfactory fusion outcomes. PAIF is capable of handling noise-related degradations, but it tends to blur the structures and details in the scene, resulting in suboptimal results. Text-IF can address illumination degradation in visible images, as well as low-contrast and random noise in infrared images, but it is ineffective against other common degradations. In contrast, our DSPFusion is able to consistently synthesize impressive fusion results across all degradation conditions. This is attributed to the fact that our degradation prior embedding network can adaptively identify degradation types, and the semantic prior diffusion model effectively recovers high-quality semantic priors. The degradation priors and high-quality semantic priors complement each other, jointly guiding the restoration and fusion model.

Table 6 illustrates the computational efficiency of different pre-enhancement algorithms. From the results, we can find that some pre-enhancement algorithms, such as Spadap, IAT, and WDNN, are computationally efficient, while others, like Hi-Diff, NeRD-Rain, and QuadPrior, introduce heavy computational burdens. In particular, QuadPrior incurs significant computational costs as it conducts the diffusion process in the image domain. Our semantic prior diffusion model recovers high-quality

Table 6: Computational efficiency of pre-enhancement algorithms.

| Task | Deblurring | Deraining | Denoising | Low-light enhancement | Exposure correction | Stripe noise remove | Average |
|---|---|---|---|---|---|---|---|
| Method | Hi-Diff | NeRD-Rain | Spadap | QuadPrior | IAT | WDNN | |
| Parm. (M) | 24.152 | 22.856 | 1.084 | 1313.39 | 0.087 | 0.013 | 226.93 |
| Flops (G) | 529.359 | 693.649 | 81.875 | 3473.413 | 6.728 | 1.105 | 797.688 |
| Time (s) | 0.359 | 0.299 | 0.003 | 1.745 | 0.007 | 0.003 | 0.403 |

semantic priors in a compact latent space, which greatly conserves computational overhead. We employ task-specific SOTA image enhancement methods for pre-enhancement, rather than relying on general approaches. On the one hand, general methods cannot simultaneously handle degradations in both infrared and visible modalities. On the other hand, general methods exhibit poor generalization on the infrared and visible image fusion datasets.

