# OpenReview forum: "DSPFusion: Degradation and Semantic Prior Dual-guided Framework for Image Fusion"
_ICLR.cc/2025/Conference — ICLR 2025 Conference Withdrawn Submission_

### Official Review · Reviewer_vDXx · 2024-10-31

**Soundness:** 2
**Presentation:** 2
**Contribution:** 3
**Rating:** 5
**Confidence:** 4

**Summary:**

This paper presents a restoration and fusion framework that utilizes both degradation and semantic priors to handle various degradation types effectively. Extensive experiments conducted in both normal and degraded scenarios demonstrate the method's effectiveness in degradation recovery and fusion. Additionally, the authors provide multiple analyses from different perspectives to validate the proposed method's effectiveness.

**Strengths:**

1. The authors design a framework that integrates degradation and semantic priors, adapting to different types of degradation. Extensive experiments confirm the effectiveness of the fusion results.
2. Various evaluation methods, including downstream tasks (such as detection), efficiency comparisons, and assessments using MLLM, provide a multidimensional evaluation.
3. The authors also analyze some failure cases, further ensuring the completeness of the method.

**Weaknesses:**

1. The illustration of the method in Figure 1 is unclear. In (a), different modules have different colors, whereas (c-e) use color-related modules, but the names are inconsistent, leading to potential confusion. It is recommended that the authors align the names of different modules. Additionally, many abbreviations in the method, such as DPEN and PGFM, are not labeled in the figure, making it difficult to follow.

2. The overall model architecture employs the diffusion guidance design from [1, 2]. The innovation in this paper lies in extending this to multi-image fusion and designing various modules. However, the authors do not sufficiently demonstrate the effectiveness of these modules, such as the proposed PGFM and SPIM.

   [1] Difﬁr: Efﬁcient diffusion model for image restoration.

   [2] Hierarchical integration diffusion model for realistic image deblurring.

3. The authors do not fully explain the settings of each experiment in Tables 1, 2, 3, and 4 (though limited space in the main text may prevent this). It is suggested to add these explanations in supplementary materials for better clarity.

4. A minor suggestion is to increase the size of images in visual comparisons, such as in Figures 3, 5, and 7 (including the enlarged patches), to improve readability.

**Questions:**

1. Please provide a more detailed introduction of the model structure, including each module's specific names (and abbreviations) in Figure 1.
2. Further analyze the effect of each design, such as PGFM and SPIM.
3. Provide details on the specific settings of each task (e.g., Table 1, Table 2, Table 3, Table 4).

---

### Official Review · Reviewer_CDUq · 2024-11-03

**Soundness:** 3
**Presentation:** 3
**Contribution:** 3
**Rating:** 5
**Confidence:** 3

**Summary:**

The paper introduces DSPFusion, a novel framework designed to enhance image fusion capabilities specifically for degraded images captured in challenging conditions. The framework leverages both degradation priors and high-quality scene semantic priors, which are restored using diffusion models. DSPFusion is noted for its efficiency, claiming a speed advantage of over 200 times compared to existing methods. The authors present a dual-prior guidance mechanism that facilitates information recovery and fusion, resulting in improved performance across various degradation scenarios.

**Strengths:**

1. The writing of this paper is easy to understand.
2. The description of the methodology is detailed.
3. The dual-prior approach is innovative, combining degradation and semantic knowledge in a novel way.

**Weaknesses:**

1. The gains brought by the proposed module in the ablation study are not significant, and the improvement observed in the ablation experiments is much smaller than the gap between DSPFusion and the current SOTA as shown in Tab.2. Additionally, these modules introduce extra parameters and computations.
2. It is difficult to be convinced that the performance of the proposed method is attributed to the introduced DPEN and SEPN. For example, the baseline without DPEN and SEPN still outperforms other methods, which is not typical of UNet performance.
3. The degradation such as noise is synthesized, and lacks visual representation in real-world scenarios.
4. The method requires training multiple networks, which is not unified.

**Questions:**

What specific types of degradation were most challenging for previous methods, and how does DSPFusion overcome these issues?

---

### Official Review · Reviewer_vaLs · 2024-11-03

**Soundness:** 3
**Presentation:** 3
**Contribution:** 2
**Rating:** 5
**Confidence:** 5

**Summary:**

This paper addresses a gap in current fusion methods, which are typically optimized for high-quality images but perform poorly on degraded images. To overcome this limitation, the authors propose a novel fusion framework designed specifically for scenarios with varying levels of image degradation. Unlike conventional approaches that rely on pre-enhancement techniques, this method directly extracts degradation and semantic priors from visible and infrared images, using them to guide the fusion process. Notably, the authors leverage a diffusion model to restore high-quality semantic priors, enhancing the fidelity of the fused images. Extensive experiments validate the effectiveness of the proposed DSPFusion, demonstrating its potential to handle degraded images more robustly than existing methods.

**Strengths:**

1. The paper is well-organized and clearly written, making it easy to follow.

2. The topic of image fusion with degraded images is highly relevant for real-world applications, addressing an interesting and necessary problem.

3. The approach of leveraging both semantic and degradation priors for image fusion is well-motivated and aligns with the needs of image fusion tasks.

4. The authors provide extensive experimental validation, including subjective quality assessments and object detection results, to demonstrate the effectiveness of their method.

**Weaknesses:**

1. The most crucial question is their ablation studies. Totally speaking, this work aims to introduce the semantic prior and degradation prior into image fusion, intending to improve the performance under different degradations. However, the ablation studies only contain the simple ablation studies by only removing DPEN and SPEN in their methods. It will cause the differences in parameters, flop. It is best to provide the comparison on computational costs. Moreover, there are lots of factors are not investigated in this work, including:

 (1) The effectiveness of the Degradation Prior Modulation Module and Semantic Prior Integration Module, Semantic Prior Diffusion Model. If you replace these module with corresponding module in Restoremore with the same parameters. Whether the proposed method is effective?

(2) The effectiveness of the proposed prior-guided fusion module is not validated. Since cross-attention is very common fusion method, It is suggested to compare simple fusion methods, like concatenation, attention, and others.









(3) The effectiveness of semantic prior diffusion module. If you substituted it with an simple enhancement module, whether it will be better? Please keep the same parameters.

(4) The effectiveness of the coefficients for different  loss function, We cannot identify which loss function is more important for this task.

(5) The effectiveness of two-stage training.

It is suggested to conduct the above ablation studies and provide the parameters and computational costs for each ablation study.

2.  Are the three t-SNE visualizations in Figure 4 are obtained at the same time? Can you give the specific configuration for each figure? Moreover, if the degradation are not observed by DPEN, how can you obtain this will not affect your proposed method?

3.  Why not only conduct a comparison in single degradation? In real-world, the degradation is usually the hybrid degradation.

4. It is not fair to compare your method with the diffusion model-based method. Actually, your proposed method cannot be treated as a diffusion-model-based method since it is not an actual generative model. Your diffusion model is only utilized to enhance the semantic prior. The whole model is actually a supervised framework like Restormer. From another perspective, please marked each parts, inlcuding (b), (c), (d), (e) in Figure 1 (a).




1.  The ablation studies are a critical area for improvement. The primary aim of this work is to introduce semantic and degradation priors to enhance image fusion performance under different degradation scenarios. However, the current ablation studies only involve removing the DPEN and SPEN modules, which introduces a reduction in parameter count and FLOPs. It would be beneficial to include a comparison of computational costs for them. Additionally, several key factors are not sufficiently explored, including:

Effectiveness of : The impact of the Degradation Prior Modulation Module, Semantic Prior Integration Module, and Semantic Prior Diffusion Model could be further clarified. For instance, replacing these modules with their counterparts from Restormer while maintaining the same parameters could demonstrate the added value of the proposed approach.

Prior-Guided Fusion Module: The effectiveness of the proposed prior-guided fusion module is not fully validated. Since cross-attention is a widely used fusion method, it would be informative to compare simpler alternatives like concatenation, standard attention mechanisms, and others.

Semantic Prior Diffusion Module: It would be useful to investigate the impact of the Semantic Prior Diffusion Module by substituting it with a simpler enhancement module, keeping parameters consistent to allow for fair comparison.

Loss Function Coefficients: The contribution of each loss function to the task is unclear. It would be valuable to identify which loss functions are most significant.

Two-Stage Training: The effectiveness of the two-stage training approach could also benefit from additional validation.

Conducting these ablation studies and reporting parameters and computational costs would greatly enhance the comprehensiveness of the analysis.

t-SNE Visualizations: Are the three t-SNE visualizations in Figure 4 obtained at the same point during training? Could you provide the specific configuration settings for each visualization? Additionally, if DPEN is unable to detect certain degradations, how would this impact the proposed method?

Single vs. Hybrid Degradation Comparison: Why not perform comparisons on single degradation types? In real-world scenarios, degradations are often hybrid in nature, so including single-degradation cases could further clarify the model’s performance scope.

Comparison with Diffusion Model-Based Methods: Comparing your approach with diffusion model-based methods may not be entirely fair, as the proposed method is not strictly diffusion-based but rather uses the diffusion model solely for semantic prior enhancement. The overall approach resembles a supervised framework like Restormer. It would also help to label each part in Figure 1(a), including (b), (c), (d), and (e), for clearer component distinction.


1. Critical Issue with Ablation Studies: The core objective of this work is to incorporate semantic and degradation priors into image fusion to enhance performance under various degradation scenarios. However, the ablation studies only involve removing DPEN and SPEN modules, which fails to capture the true impact of each component. Additionally, this ablation study will result in discrepancies in parameter count and FLOPs. It cannot identify whether the parameters cause the poor performances. Several key factors that could significantly affect the results are insufficiently explored, casting doubt on the reliability of the conclusions:

         i) Effectiveness of Individual Modules: The impact of the Degradation Prior Modulation Module, Semantic Prior Integration Module, and Semantic Prior Diffusion Model should be better validated. For instance, could these modules be replaced with their counterparts from Restormer while keeping parameters identical to demonstrate any genuine advantage of the proposed method?

         ii)  Effectiveness of Prior-Guided Fusion Module: The effectiveness of the proposed prior-guided fusion module has not been fully validated. Given that cross-attention is a widely used fusion technique, it is essential to compare it with simpler fusion methods such as concatenation or basic attention mechanisms.

        iii) Effectiveness of Semantic Prior Diffusion Module: The usefulness of the Semantic Prior Diffusion Module could be questioned. Replacing it with a simpler enhancement module under the same parameter constraints would reveal if it truly is effective.

         iv) Importance of Loss Function Coefficients: The contribution of each loss function remains unclear. Experimental validation is necessary to determine which loss function is most impactful for the task.

         v) Effectiveness of Two-Stage Training: The contribution of two-stage training also warrants further investigation.

      Overall, conducting a more comprehensive ablation study and reporting parameter counts and computational costs for each configuration are crucial for substantiating the proposed method’s claims.

2. t-SNE Visualizations: Are the three t-SNE visualizations in Figure 4 obtained under identical conditions? Please clarify the configuration for each figure. Furthermore, if the degradation is not observed by DPEN (out-of-domain), how would this affect the proposed method?

3. Why only conduct the comparison in single degradation? In the real world, the degradation is usually hybrid degradations.

4. Comparing your approach with diffusion model-based methods may not be entirely fair, as the proposed method is not strictly diffusion-based but rather uses the diffusion model solely for semantic prior enhancement. The overall approach resembles a supervised framework like Restormer. It would also help to label each part in Figure 1(a), including (b), (c), (d), and (e), for clearer component distinction.

**Questions:**

1. My main concern is the effectiveness of the proposed modules in this work.
2. The second concern is the technique contribution. Why diffusion model-based semantic enhancement is necessary?
3. If you optimize your module with hybrid degradations, it can be applied in more complex scenarios?

---

### Official Review · Reviewer_nVAJ · 2024-11-04

**Soundness:** 3
**Presentation:** 3
**Contribution:** 2
**Rating:** 3
**Confidence:** 4

**Summary:**

This paper introduces a dual-guided framework for infrared-visible image fusion, utilizing degradation and semantic priors to achieve high-quality results. The framework incorporates a latent diffusion model to enhance the quality of semantic priors.

**Strengths:**

1. The proposed method effectively addresses common degradations within a unified model.
2. The quantitative results are promising.

**Weaknesses:**

1. The design of the restoration and fusion network is overly complex, and the accompanying ablation studies are insufficient.
2. The symbols and explanations in the method section are unclear and need clarification.
3. Figure 1 does not clearly illustrate the two parallel branches, and its caption is too brief.
4. The computational efficiency of latent diffusion models, as discussed in LDM[1], is not an original contribution of this paper.
5. The training process of the diffusion model is very strange, involving first running the denoising process to obtain generated latents (semantic priors in this work) before evaluating the loss function on these generated results.
6. The qualitative comparisons presented in Figure 3 lack clarity.
7. The output of diffusion model is not shown in Figure 1 (a).

[1] Rombach R, Blattmann A, Lorenz D, et al. High-resolution image synthesis with latent diffusion models[C]//Proceedings of the IEEE/CVF conference on computer vision and pattern recognition. 2022: 10684-10695.

**Questions:**

No more questions.

---

### Official Review · Reviewer_DGZz · 2024-11-11

**Soundness:** 2
**Presentation:** 3
**Contribution:** 2
**Rating:** 5
**Confidence:** 5

**Summary:**

The article proposes DSPFusion, a dual-guided framework for image fusion that combines degradation and semantic priors to improve the quality of fused images, particularly those captured in degraded conditions. DSPFusion introduces a unique approach by utilizing both degradation-aware and semantic-driven guidance to restore and integrate information from degraded images effectively. Key innovations include a dual-prior guidance module, a semantic prior diffusion model for restoring high-quality scene context in a latent space, and a degradation prior embedding network (DPEN) that adaptively identifies and counteracts different degradation types. Through extensive experiments on multiple datasets, DSPFusion demonstrates superior performance over state-of-the-art methods, particularly in degraded scenarios.

**Strengths:**

1. DSPFusion uniquely integrates both degradation and semantic priors, offering a robust framework that enhances image quality by addressing common image degradation issues like noise, blur, and low-light conditions.
1. The semantic prior diffusion model operates in a compact latent space, which significantly reduces computational overhead compared to traditional diffusion models, achieving a two orders of magnitude improvement in processing speed.
1. The paper provides detailed experimental results on multiple benchmarks and degradation scenarios, including comparisons with state-of-the-art methods. DSPFusion’s ability to handle a wide range of degradations with minimal pre-processing is well-supported by the presented data.

**Weaknesses:**

1. The training process involves two separate stages, each with specific objectives and loss functions, which could be complex to implement and tune.
1. The choice of using degradation and semantic priors exclusively, rather than exploring alternative types of priors, is not sufficiently justified. A comparison or rationale for why these specific priors were chosen over others would help clarify their unique relevance to image fusion.
1. The paper lacks a clear explanation for employing a diffusion model specifically for degradation restoration. While diffusion models are known for their generative capabilities, a detailed comparison with other restoration approaches would help illustrate the distinct advantages of diffusion in this context.
1. Important comparative methods, such as Tardal (CVPR 2022), and CDDFuse (CVPR 2023), are omitted from the experimental benchmarks. Including these recent techniques would provide a more comprehensive evaluation of DSPFusion’s performance relative to other state-of-the-art fusion models.
    [1] Target-aware Dual Adversarial Learning and a Multi-scenario Multi-Modality Benchmark to Fuse Infrared and Visible for Object Detection. CVPR 2022
    [2] CDDFuse: Correlation-Driven Dual-Branch Feature Decomposition for Multi-Modality Image Fusion. CVPR 2023
1. The evaluation lacks several common fusion performance metrics, such as SD, SF, AG, and SSIM. Including these metrics would provide a fuller picture of DSPFusion's effectiveness and allow for a more detailed quantitative assessment across different aspects of fusion quality.
1. The experiments on degraded scenarios with enhancement might be biased, as some comparison methods lack image restoration modules. This potentially puts DSPFusion at an advantage, making the comparison unfair. A fairer evaluation would either provide these comparison methods with equivalent restoration capabilities or exclude pre-enhancement from the experimental design.
1. The model performs well under single or simple degradation types but struggles when encountering compound degradations, as noted in the discussion. When multiple degradations affect a single modality, DSPFusion’s degradation prior module may prioritize only the dominant degradation. This limitation could hinder the model’s performance in environments with complex or layered degradation effects, such as poor lighting combined with severe noise or blurring.

**Questions:**

Please refer to the relevant details in Weaknesses and rebuttal accordingly.

---

### Note · Authors · 2024-11-14

I have read and agree with the venue's withdrawal policy on behalf of myself and my co-authors.